# NIPAm-Based Modification of Poly(L-lysine): A pH-Dependent LCST-Type Thermo-Responsive Biodegradable Polymer

**DOI:** 10.3390/polym14040802

**Published:** 2022-02-18

**Authors:** Aggeliki Stamou, Hermis Iatrou, Constantinos Tsitsilianis

**Affiliations:** 1Department of Chemical Engineering, University of Patras, 26500 Patras, Greece; aggestamou@hotmail.com; 2Department of Chemistry, University of Athens, Panepistimiopolis, Zografou, 15771 Athens, Greece; iatrou@chem.uoa.gr

**Keywords:** poly(L-lysine), N-isopropylacrylamide, aza-Michael addition reaction, LCST, thermo-responsive, pH-responsive, biodegradable polymer

## Abstract

Polylysine is a biocompatible, biodegradable, water soluble polypeptide. Thanks to the pendant primary amines it bears, it is susceptible to modification reactions. In this work Poly(L-lysine) (PLL) was partially modified via the effortless free-catalysed aza-Michael addition reaction at room temperature by grafting N-isopropylacrylamide (NIPAm) moieties onto the amines. The resulting PLL-g-NIPAm exhibited LCST-type thermosensitivity. The LCST can be tuned by the NIPAm content incorporated in the macromolecules. Importantly, depending on the NIPAm content, LCST is highly dependent on pH and ionic strength due to ionization capability of the remaining free lysine residues. PLL-g-NIPAm constitutes a novel biodegradable LCST polymer that could be used as “smart” block in block copolymers and/or terpolymers, of any macromolecular architecture, to design pH/Temperature-responsive self-assemblies (nanocarriers and/or networks) for potential bio-applications.

## 1. Introduction

Polymeric “smart” materials can respond to external stimuli such as pH, light, temperature, enzymes, and electric or magnetic fields. Therefore, they are applicable in various scientific fields [1]. Thermosensitive polymers are a sub-category of smart polymers that attract great scientific interest, as they can respond to temperature and change their physicochemical properties. There are two main types of thermo-sensitive polymers. The first category includes polymers that exhibit lower critical temperature behaviour (LCST), while the second one consists of polymers that exhibit upper critical solution temperature (UCST) behaviour. The critical temperatures below and above which the polymer and the solvent become fully miscible are known as the LCST and UCST, respectively [2,3]. They change their conformation either upon heating (LCST) or cooling (UCST) from a rather random coil form to a collapsed, more globular one. Thermo-responsive polymers, integrated in block copolymers, can be exploited in a plethora of water dispersive self-assembling entities such as micelles, polymersomes, microcapsules and microgels, along with injectable hydrogels and thin films, all of them advantageous for biomedical applications such as controlled drug delivery, stem cell transplantation, wound healing, etc. [4].

Water soluble polymers that exhibit LCST are by far the most studied polymers. Among them, Poly(N-isopropylacrylamide) (PNIPAm) has attracted major attention thanks to its LCST of approximately 32 °C, which is close to and below body temperature (37 °C), which renders it suitable for biomedical applications [5,6,7,8,9,10,11]. However, provided that it is not entirely non-toxic and, more importantly, it is not biodegradable, PNIPAm does not seem to be acceptable for real biomedical applications and has mainly been used as a LCST thermo-responsive model polymer.

Polylysine is a biocompatible, biodegradable, water soluble polypeptide. These properties render polylysine and its derivatives ideal for a variety of biomedical applications [12,13]. Polylysine consists of either L-lysine or D-lysine. “L” and “D” refer to the chirality of central lysine carbon. Thus, two polypeptide isomers can be distinguished: poly-L-lysine (PLL) and poly-D-lysine (PDL) [14]. In every case, it is a weak, cationic polyelectrolyte with a pK_a_ = 10.5, thanks to the pendant free amino group that can be charged by protonation at pH < pK_a._ The polylysine conformational states in aqueous solutions, which determine its secondary polypeptide structure, have been extensively studied and are effected by a wide range of solution conditions such as pH, temperature and salt concentration [15]. In salt-free solutions and at a neutral pH, polylysine adopts a random coil conformation due to its protonated, charged state. At pH 11.5 the uncharged polylysine is entirely transformed into α-helix conformation and, upon heating, the α-helix structure is transformed into a β-sheet structure [16].

Another interesting property of poly(L-lysine) is that the free amino pendant groups it bears are susceptible to modification reactions. The objective of the present study is to combine the properties of poly(L-lysine) polypeptide with PNIPAm to form a biodegradable, LCST-type thermo-responsive polymer, in order to ultimately utilize it in real biomedical applications. The free-catalysed aza-Michael addition reaction seems suitable for this purpose. This reaction constitutes a subcategory of Michael addition reactions, a very important reaction in organic chemistry. An amine group is a Michael donor, while α, β-unsaturated carbonyl compounds act as Michael acceptors, resulting in the formation of a Michael adduct [17]. Since the amino group can act as both a base and a nucleophile, the aza-Michael addition reaction does not require the involvement of a base [18]. Michael donors can be both secondary amines and primary amines and the reaction activity depends, to a large extent, on the electronic and stereochemical environment of the amine. When a Michael donor is added to a primary amine, bis-addition is possible, depending on the added amount [19]. The choice of solvents is critical for Michael addition reactions because aprotic solvents may cause incomplete alkylation of the amino group [17]. Michael addition reaction is an effective method for modifying an existing polymer. Li and Feng synthesized a series of highly sensitive stimuli-responsive polysiloxanes (SPSis) through a catalyst-free aza-Michael addition reaction. Particularly, Poly(aminopropylmethylsiloxane) was post-modified with Nipa monomers [20].

In the present communication, preliminary results dealing with the partial modification of PLL by NIPAm monomers, denoted as PLL-g-NIPAmX (where X is mol% of NIPAm with respect to LL monomer units), via the effortless aza-Michael reaction and their resulting thermo-responsiveness are demonstrated. The modified polypeptide exhibited LCST-type thermo-sensitivity. The LCST was found to be inextricably dependent on the amount of NIPAm grafted, along with the pH and salinity of the aqueous media.

## 2. Materials and Methods

### 2.1. Materials

PLL was synthesized by ring opening polymerization of the ε-tert-butyloxycarbonyl-L-lysine-NCA (NCA: N-carboxy anhydride) monomer, followed by acid hydrolysis to deprotect the amine groups, according to standard procedures. The molecular weight characteristics of the protected PLL, determined by gel permeation chromatography (GPC, Milford, MA, USA) are M_w_ = 11,980 Daltons, M_n_ = 11,365 Daltons (degree of polymerization 50) and molecular polydispersity, M_w_/M_n_ = 1.054. Details of the synthesis and characterization of PLL and its NCA protected monomer are reported in the Appendix A. NIPAm monomers (NIPAM, Fluorochem, Derbyshire, UK) were used as received.

### 2.2. Modification of PLL by Aza-Michael Addition

PLL was modified by grafting onto the NIPAm monomers, according to Figure 1.

NIPAM monomers were added to the PLL aqueous solution (C_p_~0.1 wt%), the pH of which was adjusted close to 12, by NaOH, ensuring non-protonation of the pendant primary amines. The reaction was performed at room temperature in a flask with a ground glass stopper in different NIPAm/Lysine mol ratios and various periods of time, as discussed. The next step was the purification of the solution from unreacted NIPAm monomers and from any other impurities, through dialysis membrane (MWCO 3500 Da) with 3D-H_2_O. The final product was obtained in solid form by freeze drying. The NIPAm percentage (mol%) of the partially modified PLL-g-NIPAm samples was characterized by ^1^H NMR in deuterated water (D_2_O), using A Bruker Avance Iii Hd Prodigy Ascend Tm 600 MHz spectrometer (Billerica, MA, USA). A characteristic ^1^H-NMR spectra is presented in Appendix A.

### 2.3. Preparation of Samples for Light Scattering

Aqueous solutions of the modified PLL-g-NIPAm were prepared at a concentration of 1% *w*/*v*. The pH value of the system was adjusted by adding HCl or NaOH solution (1M). The study was performed at different pH, as PLL is pH responsive. NaCl was added to study salt effect. The concentration of the salt that forms by the addition of HCl or NaOH is negligible with respect to the concentration of added NaCl and was not considered in the final salt concentration.

### 2.4. Light Scattering

A physicochemical study of the modified PLL-g-NIPAm was performed by the static light scattering (SLS) technique. All measurements were carried out using a thermally regulated spectro-goniometer, Model BI-200SM (Brookhaven Holtsville, NY, USA), equipped with a He–Ne laser (632.8 nm). The aim of the study was to determine whether the modified polymers were thermo-sensitive and, in this case, to identify their critical temperature, T_LCST_. More specifically, in a PLL-g-NIPAmX sample (transparent aqueous solution at low temperature), the light scattered intensity (*I*) was measured at a constant angle θ (90°) by increasing, stepwise, the temperature. At each temperature, sufficient time was left for equilibration before three repeating measurements were taken. The T_LCST_ was determined at T, above which an abrupt increase in LS intensity was observed. For the sake of comparison, the temperature-dependent optical density (OD) was also measured for one sample by UV visible to detect the cloud point T_cp_, i.e., the temperature above which the solution turns turbid, using a Hitachi U-2001 UV–VIS spectrophotometer (Schaumburg, IL, USA).

## 3. Results and Discussion

The critical temperature of the aqueous solutions was determined by static light scattering which, in fact denotes, the onset of association (aggregation), T_ass_, of the macromolecules, which is the consequence of their hydrophilic to hydrophobic transition following the phase separation of the solution upon heating. The T_ass_ coincides with the cloud point, T_cp_, as observed by optical density measurements (Figure 1), since both are due to the intermolecular hydrophobic association. To emphasise that critical temperature occurs upon heating, we will refer to it hereafter as T_LCST_.

Initially, the thermal behaviour of pure PLL in aqueous solutions was investigated at high pH (10.6), where the PLL is nearly uncharged (bearing non-protonated pendant primary amines), as it is shown by z-potential measurements [16]. As can be seen in Figure 2A, the light scattering intensity increases gradually above 45 °C, denoting the onset of association, and levels above 75 °C. On the contrary, no temperature effect is detected at pH 7. At pH 10.6, the 1 wt% polymer solution remains transparent in the entire temperature range investigated. However, a free-standing gel forms above 65 °C, as seen in Figure 2A. As it is well known, PLL adopts various secondary structures depending on pH and temperature [15,21]. At pH 10.6, the nearly uncharged PLL adopts mainly an α-helix secondary structure, i.e., about 80%, which becomes 100% above pH 11.5 [22]. Upon heating an *α*-helix-to-*β*-sheet conformational transition is expected above a critical temperature which depends on molecular weight, i.e., the lower the molecular weight the higher the transition temperature [23]. The thermo-induced formation of the transparent gel observed in Figure 2A is attributed to the *α*-helix-to-*β*-sheet transformation of the majority of the PLL chains, associated with the formation of a spanning 3D physical network. As reported, the peptides, when adopting a *β*-sheet secondary structure, self-assemble, forming entangled fibrils, which justifies the gel appearance [24]. The transition temperature was detected at about 45 °C, which seems reasonable for the molecular weight of PLL (Mn = 6500), as it is known that it depends on PLL molecular weight [25]. At pH 7, PLL does not exhibit thermosensitivity, since it exists as a random coil, due to its charged state.

Partial modification of PLL by conjugating NIPAm moieties onto primary amines via the aza-Michael addition reaction, with a feed molar ratio [NIPAm]/[LL] of 35 mL%, was accomplished first. The reaction was carried out at room temperature. The yield of the reaction was 82%, meaning that the percentage of NIPAm moieties incorporated into the PLL chain was 28.6 mol% (denoted as PLL-g-NIPAm29). The scattering intensity as a function of temperature in aqueous solutions of the partially modified PLL-g-NIPAm29 is shown in Figure 2B at various pH 7.8, 8.7 and 10.8. At pH = 7.8 and 8.7, the solutions are clear, and the intensity remains constant upon increasing temperature. The modified polymer does not show any thermo-sensitivity under these conditions. This behavior is attributed to the fact that PLL is positively charged, as the free primary amine groups are protonated (pH < pKa). At these pH conditions, the sample under study is characterized, to a large extent, by hydrophilicity, exhibiting very good solubility in the aqueous medium. Therefore, the grafting of NIPAm groups at 29%, does not appear to affect the thermal responsiveness of the samples, investigated in this temperature range, provided that the percentage of positively charged amine groups predominate. At pH 10.8, the solution shows thermosensitivity as the intensity of the light scattering increases abruptly above 54 °C. The presence of the NIPAm groups on the chain generate two new effects on the heat-induced solution behaviour. The scattering light intensity increases more sharply and the solution exhibits turbidity at higher temperatures. These effects are consistent with LCST-type thermo-sensitivity. Remarkably, at 60 ^ο^C the solution was transformed to a free-standing gel. It seems probable that the low percentage (29 mol%) of NIPAm attachment on the PLL chains does not entirely prevent the *α*-helix-to-*β*-sheet conformational transition, inducing gelation.

By increasing the [NIPAm]/[LL] feed ratio to 78%, the percentage of NIPAm modification reached 69 mol% (PLL-g-NIPAm69). The PLL-g-NIPAm69 aqueous solution at pH 11.8 exhibits a sharp increase in the light scattering intensity at 32 °C, above which the solution turns turbid, as shown in Figure 3A. More importantly, no gel formation was observed (Figure 3A, inset), implying that, at this NIPAm percentage, the formation of *β*-sheet responsible for the gel formation is prevented. Therefore, the observed thermal transition is clearly the LCST type. By decreasing pH at 9.5 and 8.3, the transition disappears at least up to 80 °C, due to ionization of the remaining free amine groups. Provided that the charged groups are highly hydrophilic, the T_LCST_ is likely shifted to higher temperatures.

The effect of ionic strength was explored by adding NaCl at various pHs, in an attempt to decrease the LCST. In Figure 3B, we can see the effect of pH in saline aqueous solutions of 1M NaCl. The thermosensitivity appears at T_LCST_ = 30 °C in pH 9.6, whereas it is not visible at lower pHs up to 80 °C. Compared to the absence of transition at pH 9.5 in salt free solutions (Figure 3A), the presence of 1M salt has a remarkable effect on the LCST shift, greater than 50 °C. At the slightly higher pH of 9.8 the critical temperature continues to decrease significantly to T_LCST_ = 14 °C. These intensive effects should be attributed to the partially substituted PLL with 69 mol% NIPAm which still bears a high number of free amines that are ionizable, depending on pH. Thus, the increase in ionic strength likely exerts two effects. It is known that hydrophilic comonomers increase the LCST, whereas hydrophobic ones have the opposite effect [4]. The presence of salts decreases the hydrophilicity of the protonated amines by electrostatic screening, thus inducing a decrease in LCST. Moreover, the well-known salting out effect of NIPAm moieties operates in the same direction [4,26]. Hence, both effects seem to be the reason for the observed LCST high shift at pH 9.5 for the PLL-g-NIPAm69 sample. At pH 9.0, more salt is needed, e.g., 1.5 M, for the appearance of thermosensitivity, which is detected at T_LCST_ = 55 °C. Finally, at pH 8.8/2M NaCl, the T_LCST_ is equivalent to that at pH 9.8/1M NaCl, clearly showing the combined effect of pH and ionic strength.

By increasing the percentage of NIPAm integration to 87 mol% (PLL-g-NIPAm87), the T_LCST_ is further shifted to lower temperatures, as seen in Figure 4. For the uncharged state (high pH) the T_LCST_ drops to 27 °C versus the 32 °C of 69 mol% content. At pH 9 and salt free solution it appears at 50 °C in contrast to the PLL-g-NIPAm69 sample, in which thermosensitivity is not visible, even up to 80 °C (Figure 3B). At the lower pH 8, a weak thermosensitivity appears at 65 °C.

For the LCST to approach physiological conditions, namely pH 7.4, 37 °C, the following “in situ” experiment was designed (Figure 2). To an aqueous solution of PLL-g-NIPAm87, a successive addition of NIPAm monomers was applied. Specifically, to 20 mg (0.095 mmol) of PLL-g-NIPAm87 an amount of 0.057 mmol of NIPAm was added to 2 mL aqueous solutions. NaOH (1M) was also added to adjust the pH of the mixture at 12. The solution was left under stirring for 48 h, and product 1 was characterized in terms of thermosensitivity at various pH. The same procedure was repeated sequentially three times, as demonstrated in Figure 2. The final product was characterized by ^1^H-NMR (i.e., PLL-g-NIPAm116). The successive addition of NIPAm resulted in an ongoing insertion of this moiety into the remaining free amines of PLL, reaching, after four steps, a total NIPAm percentage of 116 mol%, which is higher than 100%. This is attributed to the fact that the secondary amine remaining after the first addition is more nucleophilic than the primary one and bis-addition is unavoidable [20]. As can be seen in Figure 5, for the uncharged samples (pH > 11, red symbols), the consequence of NIPAm integration imposes a continuous T_LCST_ shift to lower temperature, about 2 or 3 degrees after each NIPAm addition, finally reaching 17 °C. More importantly, the T_LCST_ appears at lower pHs as the percentage of the remaining ionizable free amines decreases. For instance, in the PLL-g-NIPAm116 (Figure 5, product 4), the T_LCST_ appears at pH 7.4 at 39 °C, while it is not visible up to 55 °C when the pH decreased to 6.9, indicating that a considerable number of free amines remains intact, and their ionization capability still affects the LCST. It can also be observed that bis-addition is greater than 16%.

By attempting to increase the NIPAm content in a single experiment, the aza-Michael addition reaction was accomplished in NIPAm/LL ratio 2/1, prolonging the time at room temperature. The content of NIPAm reached 98 mol%, likely showing steric hindrance effects. It seems that the successive addition of NIPAm of the “in situ” experiment could enrich the PLL with more NIPAm (116 mol%). The temperature dependence of the light scattering intensity of the 1% PLL-g-NIPAm98 aqueous solutions at various pH and salt concentrations is depicted in Figure 6a. In the uncharged state of LL moieties (pH 11), the T_LCST_ was observed at 18 °C, which is shifted slightly at 20 °C when the pH dropped to 9. However, at pH 7.5, the intensity did not change upon the increase in temperature, at least up to 60 °C, due to ionization of the free remaining amines. Upon adding 0.15M NaCl, the T_LCST_ appeared again at 52 °C, which further decreased with increasing NaCl concentration (Figure 6b). Finally, when 0.6M NaCl was added, the T_LCST_ appeared at 37 °C.

By extrapolating to zero salt concentration, the T_LCST_ should appear at 57 °C in pH 7.5, which is not valid, as shown by the light scattering data in Figure 6a. Moreover, the T_LCST_ shift with salt concentration (ΔT more than 20 °C) is much more pronounced compared with the behaviour of pure PNIPAm (ΔT about 4 °C) in the presence of 0.6M NaCl, attributed to the salting out effect [26]. Therefore, the strong salt effect is clearly due to the additional electrostatic screening of the charged amine groups that decrease their hydrophilicity, as discussed previously.

The T_LCST_ data of the various PLL-g-NIPAm samples is summarized in Figure 7. The influence of the grafted NIPAm percentage at high pH > 11, where the remaining free amines are in an uncharged state, is depicted in Figure 7a. The data follow a decreasing linear dependence of T_LCST_ with the mol% NIPAm, since the continuous decrease in the remaining amount of the hydrophilic primary amine groups also decreases the LCST.

More importantly, for the partially grafted samples, T_LCST_ is strongly pH-dependent. As observed in Figure 7b, the T_LCST_ decreases with increasing pH and levels off at high pH. Moreover, the degree of NIPAm grafted (mol%) remarkably influences this dependence; that is, the higher NIPAm content the lower T_LCST_. For instance, by increasing the NIPam content from 87 to 116 mol% at pH 9, the LCST is shifted by more than 30 °C at lower values. Therefore, the PLL-g-NIPAm is in fact a pH/thermo dual-responsive polymer.

## 4. Conclusions

The target of the present work was to prepare an LCST-type thermosensitive macromolecule, based on PLL polypeptide to ensure biodegradability. For this purpose, NIPAm moieties were conjugated onto the free amine pendants of PLL, via the effortless free-catalysed aza-Michael addition reaction in aqueous media at room temperature. The reaction yield was estimated at about 80%. A series of partially modified PLL was prepared. The addition reaction gave mono- and bis-addition products with free remaining primary amines (Figure 1).

The partially modified PLL-g-NIPAmX exhibited LCST-type thermo-responsiveness, which is highly tuneable by the content of the incorporated NIPAm moieties, i.e., the LCST decreased with increasing NIPAm content of the polymer. More importantly, thanks to the presence of the remaining ionizable free primary amines, the LCST of the partially modified PLL is highly sensitive to pH and ionic strength. For the given NIPAM content, the LCST decreased with pH and salt concentration. The aza-Michael reaction between amines and double bonds can be applied to any LCST-type vinylic monomer, e.g., oligo(ethylene glycol) methacrylate, and polypeptides bearing primary and/or secondary amines, e.g., poly(L-histidine), towards biodegradable LCST-thermo-responsive polymers. It should also be mentioned that similar thermo-responsive modified polypeptides have been reported previously using different reactions [27,28,29,30,31]. The advantage of the present work relies on the simplest free-catalysed aza-Michael addition reaction, forming stable covalent bonds that can be accomplished at room temperature without any by-products and can be applied to any peptide (and/or protein) comprising amino acid residues bearing pendant amines.

The protected PLL precursor, bearing N-termini, can be integrated to block-type segmented polymers, e.g., block and/or graft copolymers, prior to deprotection and modification. These copolymers are expected to form pH/thermo dual-responsive “smart” self-assemblies that could be used as nano carriers and/or 3D networks (scaffolds). This constitutes our ongoing research, and the perspectives towards bio-applications seem promising.

## Data Availability

The data presented in this study are available on request from the corresponding author.

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
