# Peer review of "NIPAm-Based Modification of Poly(L-lysine): A pH-Dependent LCST-Type Thermo-Responsive Biodegradable Polymer"

_polymers, 2022, doi:10.3390/polym14040802_

Round 1

Reviewer 1 Report

The authors submitted their manuscript entitled “NIPAm-based Modification of Poly(L-lysine): a pH-dependent LCST-type Thermoresponsive biodegradable polymer”. In this work, Poly(L-lysine) was partially modified via Michael addicion reaction by grafting ontono N-isopropil(acreilamide). The final product exhibited LCST termosensitivity. The results obtained in this article are very interesting since LCST-type thermosensitivity depends largely on pH and ionic strength.  After reviewing the manuscript, the reviewer recommends some revisions for that the manuscript is suitable for publication in Polymers.

- The authors comment that the LCST polymer has great potential for biomedical applications. It would be interesting if the authors commented on these applications in the introduction to the paper.

- References are not correctly indicated in the text. References should be placed in square brackets (eg [1]).

- NCA is not defined in the text. It is indicated in the supplementary information (NCA: N-carboxy anhydride) but not in the paper.

- Line 105. Please, remove the black square at the end of the text.

- Line 125. For clarity, cloud point (Tcp) should be defined in the text.

- Line 131. Why does Tass coincide with the cloud point?

- The captions of the figure are not in correct format. Please, to put in the correct format according to the instructions for Polymers.

- Line 212. “the well-known salting out effect of NIPAm moieties operates in the same direction”. Are there references on this?

- In the reference section, to put in the correct format according to the instructions for Polymers.

- In the Figures 6 and 7, the indicators a and b must be written in the correct format. Authors should read the instructions for Polymers.

- The Scheme 2 with the synthesis of PLL-g-NIPAmX should not be in conclusions. This schematic should be in the materials and methods section

Author Response

We would like to thank the Reviewer for the fruitful and creative comments.

- The authors comment that the LCST polymer has great potential for biomedical applications. It would be interesting if the authors commented on these applications in the introduction to the paper.

 We have added in lines 36,37: “such as controlled drug delivery, stem cell transplantation, wound healing etc.”

- References are not correctly indicated in the text. References should be placed in square brackets (eg [1]).

 Done.

- NCA is not defined in the text. It is indicated in the supplementary information (NCA: N-carboxy anhydride) but not in the paper.

 Added in section 2.1 line 87.

- Line 105. Please, remove the black square at the end of the text.

 Done.

- Line 125. For clarity, cloud point (Tcp) should be defined in the text.

 We have added “i.e., the temperature above which the solution turns turbid.”

- Line 131. Why does Tass coincide with the cloud point?

 We have added “since both are due to the intermolecular hydrophobic association.”

- The captions of the figure are not in correct format. Please, to put in the correct format according to the instructions for Polymers.

 We have checked it.  

- Line 212. “the well-known salting out effect of NIPAm moieties operates in the same direction”. Are there references on this?

 We added ref. 4 and 26.

- In the reference section, to put in the correct format according to the instructions for Polymers.

 Done.

- In the Figures 6 and 7, the indicators a and b must be written in the correct format. Authors should read the instructions for Polymers.

Done. 

- The Scheme 2 with the synthesis of PLL-g-NIPAmX should not be in conclusions. This schematic should be in the materials and methods section

The scheme 2 transferred to section 2.2.

Reviewer 2 Report

This research is useful for a wide range. This manuscript can be accepted after minor revision.

(i)English language needs to correct throughout the manuscript.

(ii) Rewrite the conclusion by providing exact findings from the results and discussion.

Author Response

We would like to thank the Reviewer for the fruitful and creative comments. Below see our respond.  

(i)English language needs to correct throughout the manuscript.

We tried our best to improve the English language.

(ii) Rewrite the conclusion by providing exact findings from the results and discussion.

We have revised the Conclusions.